# The Realms of Abandonment: Measures and Interpretations of Landscape Value/Risk in Northern Sardinia (Italy)

Antonello Monsù Scolaro and Cheren Cappello *

Department of Architecture Design and Urban Planning, University of Sassari, Piazza Duomo 6, 07041 Alghero, Italy; amscolaro@uniss.it
* Correspondence: c.cappello@studenti.uniss.it; Tel.: +39-3899195326

**Abstract:** This contribution is part of the context of studies on the prospects of eco-oriented territorial rebalancing involving the settlement networks of inland areas. These are characterised by the contrast between socio-territorial disadvantage issues and opportunities to reuse physical resources within the broader framework of territorial regeneration and the revitalisation of local identities. In Italy, the region of Sardinia represents one of the most suitable operational contexts for the study of this relationship due to the presence of a natural context that dominates the urbanised areas and a deep, and in some ways still intact, cultural identity. Between nature and culture lies the issue of urban settlement structures, which are progressively being emptied due to depopulation and abandonment, and which require responses to revitalise territories integrated with the now inescapable ecological–environmental needs. This study proposes the formation of an initial platform of indicators to describe the effects of land abandonment through a multidimensional approach to highlight the potentials and weaknesses of the natural, urban, and socio-cultural heritage. The scale of observation and comparison concerns urban centres and small towns in the province of Sassari in the Region of Sardinia (Italy). The creation of an integrated set of maps highlighting deficiencies, vocations, and unexpressed potentials are the first results of the observation methodology adopted; these residual potentials can be used to design possible redevelopment and regeneration strategies based on the specific vocations of territories and urban settlements.

**Keywords:** culture and society; abandonment risk; strategic urban planning; land recovery

## 1. Landscapes and Interconnected Risks

The landscape is one of the most important references—in terms of its scale and complexity—for 'judging' territorial phenomena of natural and/or anthropogenic origin. This study deals with the representation of landscape risk, manifested in the various forms of abandonment, in a sub-regional territorial context strongly characterised by tensions between certain development drivers and the enduring socio-economic order still preserved in the slow and coherent evolution of settlement forms. The anthropic dimension of the landscape highlights the vocations and unexpressed potentials of territories, which can be traced back to the sphere of values based on which local disadvantages are selected; the landscape risk results from the asymmetry between territorial resources and the location of the population, in short, the degree and forms of abandonment. This aspect of landscape risk is now increasingly and irreversibly linked to the growing anthropogenic pressure resulting from lifestyles based on productive efficiency rather than the resilience of ecological and natural systems.

### 1.1. Nature and Culture of Landscapes

The concept of the landscape encompasses a wide range of knowledge, which, while contributing to different interpretations and representations, attempts to regulate the

dialectic between natural structure and anthropic transformations [1], summarised in the terms 'pristine landscape' (Urlandschaft) and 'cultural landscape' (Kulturlandschaft) [2].

Sauer, with his physiognomic approach, provides an interesting integrated interpretation: the landscape has an 'organic quality' because it is a part of the real territory with recognisable characteristics. A landscape can be defined as 'an area consisting of a distinct association of forms, both physical and cultural' [3]. The natural landscape is the landscape as it existed before human intervention. It consists of a collection of natural landforms: mountains; hills; plains; plateaus; lakes and watercourses; vegetation; and other landscape elements [4]. The cultural landscape, on the other hand, is that which has been modified by human use (agriculture and grazing), its economic exploitation (factories, industries, and production sites), or its modification (settlements and urbanity of various origins and types); in 1992, the United Nations recognised human interactions with natural landscapes as cultural landscapes. [5].

As a reference for functional classifications for protection, conservation, and development actions of settled communities, further degrees of relationship between anthropic action and the original spatial order can be taken: (1) built landscape; (2) organically developed landscape; (3) associative landscape. In 2000, the European Landscape Convention reaffirmed the perceptual component of landscape as a discriminating factor that allows us to reconstruct both the interactions between nature and living species and the deep, often hidden, processes of transformation [6,7] and to interpret their modes of interaction [8,9].

Landscape culture, understood as the ability to distinguish and combine its components in several dimensions, is embodied in a holistic view through the coupling of different landscape systems [10]. It is through these systems that man perceives and evaluates their existence and, at the same time, interacts with them by transforming and creating them [11].

The 'culture of the landscape' implies the inseparability of its material and immaterial dimensions in the progressive affirmation of a territorial identity whose character derives from the action of natural and human factors and their interrelationships [12].

### 1.2. The Environmental Issues and the Landscape Risk

Territorial dynamics, driven by the progressive evolution of lifestyles and increasing anthropogenic pressures, raise the question of the compatibility between consolidated historical forms (natural and artificial) and the current transformations required by territories in global competition; these transformations have reached such proportions that they are leading to the progressive alteration and impoverishment of the original natural and identity characteristics of landscapes [13].

The growing availability of technology has boosted humanity's capacity to modify the natural landscape, which, however, does not become the cultural landscape due to irreversible large-scale transformations with definitive consequences for the climatic and ecosystemic order [14]. The World Economic Forum's Global Risk Report 2023 points out that landscape degradation leads to diminishing natural resources, collapsing local economies, increased conflict and forced migration [15]. In Italy, as elsewhere, the uncontrolled growth of cities and their expansion into the countryside has profoundly altered the landscape, resulting in enormous land consumption, reduced biodiversity, and increased energy use [16].

Today, the processes of land use and exploitation, superimposed on the depopulation and abandonment of sites, have consequences at different scales, both urban and territorial. The combination of demographic dynamics and deindustrialisation processes has created several abandoned landscapes that still contain resources—natural and man-made—from which the regeneration and urban reconnection of these places could start, also in the broader context of the ecological transition strategy outlined in the United Nations (UN) 2030 Agenda for Sustainable Development (SDGs) [17,18].

The condition of abandonment implies the presence of a substantial endowment of unused assets whose vocations (intrinsic characteristics) and unexpressed potentials (extrinsic or contextual characteristics) redefine the ways of 'implementing urban and

territorial planning a new state of necessity capable of seizing a historic opportunity for concrete transformation' [19]. Thus, derelict goods and places offer innovative planning opportunities to reconnect pre-existing resources with people and lifestyles [20].

This implies a broader vision of a common landscape restoration plan and an awareness of embodied resources as legacies of past eras and lifestyles. From the perspective of possible cross-sectoral and multi-level cooperation between stakeholders and institutional decision-makers, interlinked landscape risks could be addressed, promoting effective interventions across institutional, political, and national boundaries and restoring productive conditions for long-term resilient areas [21].

*1.3. Contents and Aims*

This paper proposes the construction of a queryable database that, through a series of parameters on the state of the main components of the territory of Northern Sardinia (Figure 1), describes the detailed vitality/abandonment conditions of the municipal units with reference to natural, urban, and socio-cultural aspects.

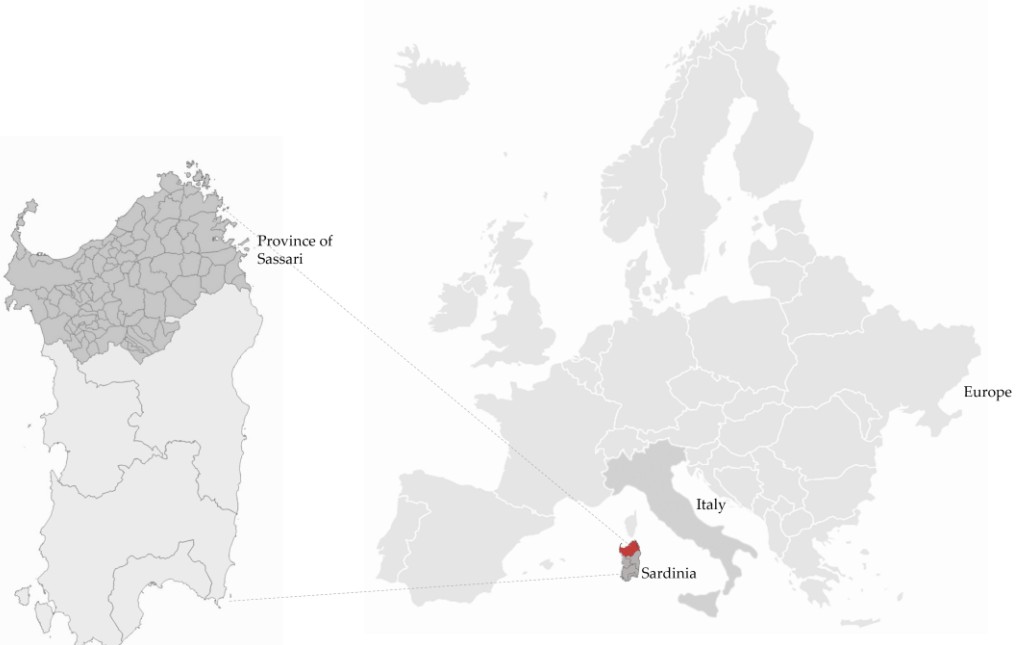

**Figure 1.** Framing of the territories under analysis. (Our processing).

Using spatial analysis tools (GIS) and starting from the official data available on the portals of the competent national institutions—public bodies and institutions that provide aggregated data for specific uses—this study provides analyses aimed at a synthetic reading of the natural, urban, and socio-cultural components of the area under study, through indices and maps at different levels of aggregation. The summary results identify criticalities and potentials that can support subsequent planning and related revitalisation actions for specific areas and urban centres [22,23].

Therefore, the title recalls the double value of the proposed methodology: on the one hand, a container of knowledge useful to represent the value of the landscape—here understood as the set of physical systems (abiotic and biotic components and built apparatuses) and flow systems (the established communities, their quality of life and their economic-productive characteristics)—the area analysed; on the other hand, a survey of landscape risk, here reduced to the concept of abandonment of structures that have modified the natural landscape but that, being without function, do not even constitute a cultural landscape. The model then provides measures of the degree of landscape value and the degree of abandonment, which, interpreted through a correlation of their different compo-

nents, provide a basis of useful information for planning rehabilitation and revitalisation at different scales.

This paper is divided into six parts: Section 2 (Materials) provides a general description of the study context in terms of the main measures of the degree of abandonment and, more generally, of the landscape's environmental, material, and socio-economic vulnerability; Section 3 (Method) describes the procedure via which the most relevant information was selected and aggregated in order to construct the most representative synthetic indices of the study context in terms of landscape risk; Section 4 (Application and Results) presents the results and the main interpretations of the natural, urban, and socio-economic contexts in an organic way; Section 5 (Discussion) discusses the results through comparisons and correlations between landscape risk and the drivers of abandonment; Section 6 (Conclusions) summarises the main lines of this study, indicates some of its limitations, and outlines the main perspectives.

## 2. Materials

Sardinia covers an area of 24,100 square kilometres and is the second-largest island in Italy. It is made up of 377 municipalities divided into four provinces, the capital of which is Cagliari.

Sardinia's geographical position, eccentric with respect to the main Mediterranean routes, has favoured the development of indigenous flora and fauna that has evolved, being protected from contamination and hybridisation [24].

For this reason, too, Sardinia has always been sparsely populated. It experienced a period of intense demographic growth between 1920 and the second post-war period, maintaining the highest birth rate in Italy, but with the beginning of the 21st century, the trend reversed [25]. Between 1991 and 2001, there was a decrease of only 2%, which shows considerable stability of the population, to some extent reproducing the Italian trend (Figure 2).

The case study of Sardinia stands out as a particular case within the European macro issue of depopulation and shrinking cities. The phenomenon appears to be rather complex, difficult to define unambiguously and interrelated with external factors that depend entirely on the context in which it occurs, taking on very different connotations and dynamics [26]. For this reason, it is very difficult to compare different methodologies, which, in order to be valid, must be based on knowledge of the specific characteristics of the reference context [27]. The effects of depopulation and the abandonment of large parts of cities draw attention to the opportunities offered by the availability of living and working spaces to be rewritten, economies to be reinvented, and social plots to be reconstructed for the formation of a new culture of urban and territorial planning that takes into account these dynamics, which are no longer unprecedented [28].

The Sardinian region, thanks to its isolation, has preserved a landscape that is still very natural and precious in the face of the crisis that more 'consumed' regions are facing. At the same time, however, the sparse population increases the risk of shrinkage, and the inadequacy of services, especially accessibility, aggravates the situation of small inland communities.

In Italy, as part of the National Reform Programme (PNR), the 2014–2020 Partnership Agreement defined the National Strategy for Inner Areas (SNAI) in 2014, with the aim of defining an organic governance framework to halt the demographic decline and relaunching the economic development of territories [29]. In total, the SNAI identifies 124 project areas covering 1904 Italian municipalities. Initially, Sardinia was included in the SNAI with two experimental areas—Alta Marmilla and Gennargentu Mandrolisai—to which two other areas—Barbagia and Valle del Cedrino—were added. The number of municipalities included in the four areas identified is 318, representing 84% of the total number of municipalities in the region [30,31].

The demographic dynamics reflect the crisis of production and the agricultural and pastoral settlement that manifested itself in the last decade of the 20th century. In fact, the population moved towards the more integrated economies of the province of Cagliari and

towards the eastern coasts, attracted by the demand expressed by the tourist sector in these areas (Figure 3) [32,33].

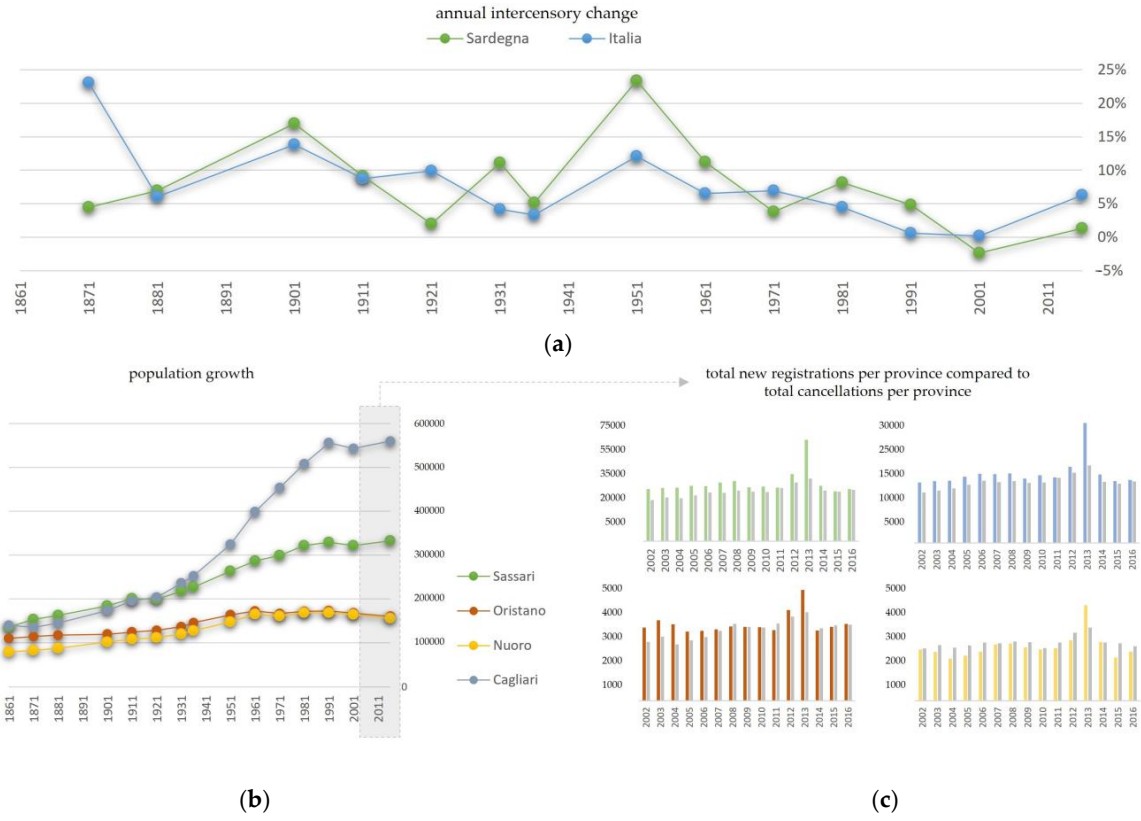

**Figure 2.** Annual intercensual variation in Italy and Sardinia (**a**); detailed analysis of the four provinces of the region of Sardinia of population growth from 1861 to 2011 (**b**); and comparison between cancellations and new arrivals by year (**c**). (Our processing).

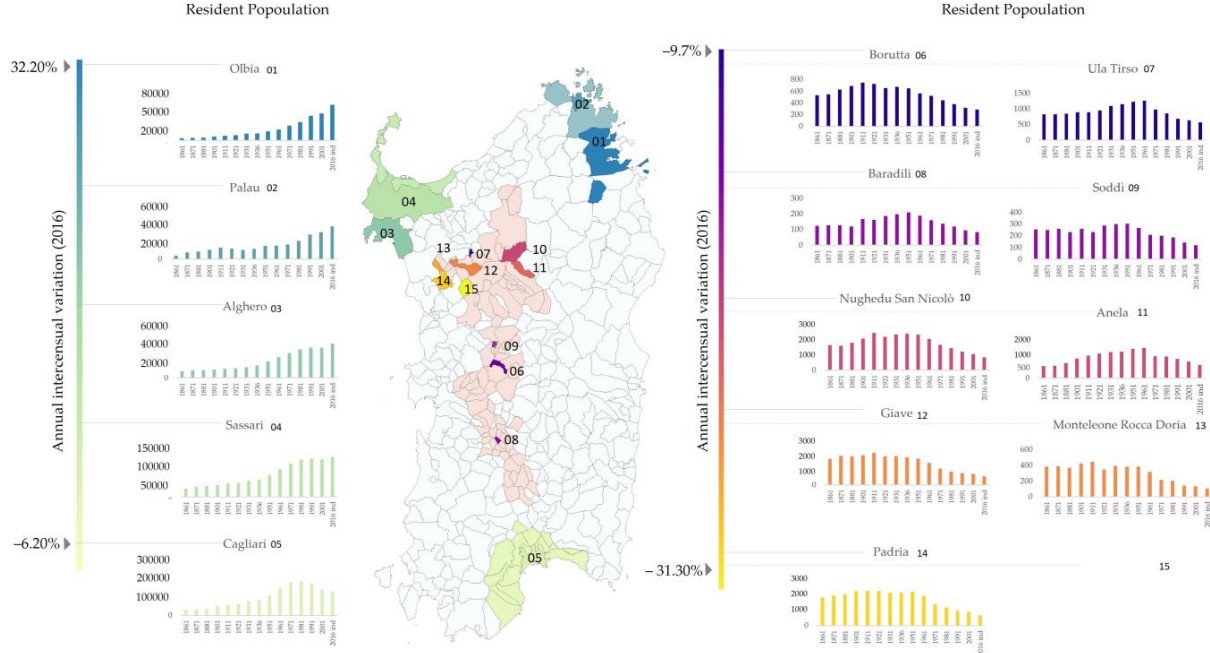

**Figure 3.** Annual intercensual variation of the most significant growing municipalities (**left**) and the municipalities most at risk of depopulation (**right**) in the region of Sardinia. (Our processing).

The economy of Sardinia today is mainly based on the tertiary sector (78% of the workforce), driven by tourism and the seaside. The primary sector accounts for 6.2% of employment and is mainly based on sheep farming. The manufacturing sector, represented by the metallurgical (Porto Torres), chemical (Ottana), and petrochemical (Olbia and Cagliari) industries, accounts for the rest of the workforce (Figure 4) [34].

| | Sardinia | | | Italy | | |
|---|---|---|---|---|---|---|
| | incidence | | % | incidence | | % |
| | 2005 | 2014 | 13-14 | 2005 | 2014 | 13-14 |
| agricolture | 6.9 | 3.9 | −7.8 | 3.8 | 3.3 | −4.6 |
| total industry | 9.6 | 19.7 | 11.6 | 25.9 | 26.8 | −2.4 |
| extractions | 0.3 | 0.5 | 47.7 | 0.5 | 0.7 | 25.2 |
| manifacture | 5.6 | 4.6 | −3.4 | 16.6 | 19.4 | 1.1 |
| energy, gas | 1.1 | 12 | 24.9 | 4.8 | 3.5 | −20.2 |
| water, waste | 0.7 | 0.8 | −7 | 1.1 | 1.1 | −5.7 |
| construction | 1.9 | 1.8 | −13 | 2.8 | 2 | −2.8 |
| total services | 83.5 | 76.3 | 2.9 | 70.3 | 69.9 | −3.2 |
| trade | 8.7 | 2.7 | -19 | 6 | 4.7 | −17.3 |
| transport | 9.1 | 5.8 | 20.8 | 7.4 | 7.4 | 0.3 |
| real estate | 36.5 | 35.4 | −5.7 | 29.6 | 31.6 | −2.7 |
| PA, compulsory insurance | 16.1 | 18 | 21.2 | 9.1 | 8.1 | −4.2 |
| other services | 13 | 14.4 | 5.6 | 18.2 | 18.1 | −0.6 |

(**a**)　　　　　　　　　　　　　　　　　　　　(**b**)

**Figure 4.** Articulation of the three economic macro-sectors and the incidences of industries in percentage (**a**) and with respect to ATECO economic activity classifications (**b**), where (A) agriculture, forestry, and fishing; (B–E) mining, manufacturing, energy, and water supply; (F) construction; (G–J) trade, transportation, accommodation and catering, information, and communication; (K–N) financial, insurance, real estate, professional, and scientific and support services; (O–T) public administration, education, health, and other. (Our processing).

After the Second World War, the 'Rebirth Plan' was devised to improve the socio-economic situation of the island, which was in a deep state of backwardness. Launched in 1962, it financed various entrepreneurial and industrial initiatives until 1975 [35]. However, the plan did not have the desired results, mainly because the establishment of petrochemical industries did not act as a driving force for the entire region, and social structures did not follow the political intentions of the plan's creators. To this day, Sardinia suffers from organisational, infrastructural, and productive difficulties due both to the wrong choices made in those years and to a socio-cultural structure that has changed little since the 1950s [36].

Local economies are unable to compete with global markets, and employment opportunities in the inland areas are diminishing; the result is the abandonment of areas and small villages. The productive landscapes of the past become landscapes of abandonment as people migrate to the coasts, where seasonal tourism attracts both inland inhabitants and new residents. In this way, the primary services of the interior disappear while those of the coast are strengthened, a region that resembles a 'crater, empty in the centre and full on the sides, like an empty shell' [37].

*Case of Study*

This study focuses on the province of Sassari, in northern Sardinia, which is the area most affected by the demographic flows and socio-economic changes described. This area has been analysed and described in terms of its natural, urban, social, and economic systems.

In 2016, the Province of Sassari acquired (Regional Law No. 2 of 4 February 2016) the territories of the municipalities of the former Province of Olbia Tempio, becoming one of the largest provinces in Italy with an area of 7692 square kilometres divided into 92 municipalities (Figure 5a). It is the most populous province in the region, with a total population of 493,788 inhabitants (Figure 5b,c).

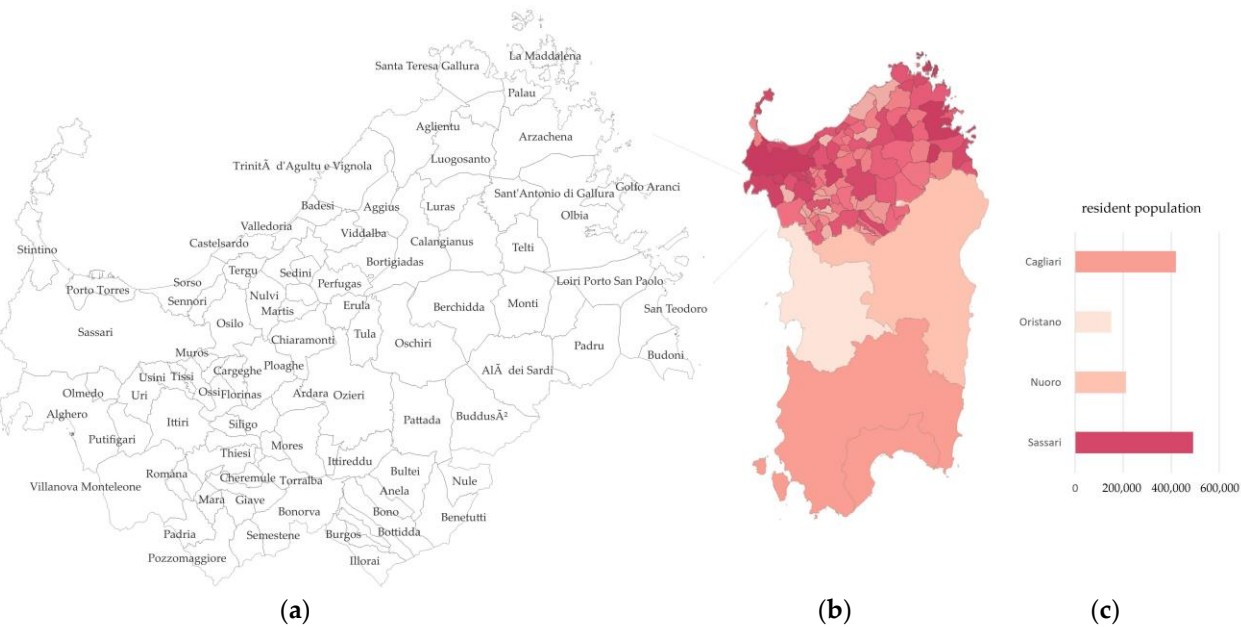

**Figure 5.** (**a**) Map of the provincial municipalities to be analysed; (**b**) ranking of the population per municipality in Sassari compared to the total of the other provinces in the area; in this representation, the average colour of Sassari province shown in the graph (**c**) is detailed by municipality based on a corresponding shade; (**c**) population of the four provinces in Sardinia. (Our processing).

This is probably due to the better quality of transport and travel services to and from the island: the province of Sassari is home to two of Sardinia's three airports (Olbia and Fertilia–Alghero) and two of its three ports (Olbia and Porto Torres).

In terms of landscape, this area is characterised by rugged coastlines with different geological features and colours, while the northeastern area is predominantly mountainous, with the famous granite reliefs of Gallura.

This part of the region, Gallura, can be considered a separate entity in economic and demographic contrast to the rest of the island and is, therefore, interesting to study. Since the 1960s, the historical region of Gallura has been internationally renowned for its cork and granite industries, which in the last two decades have become less competitive on international markets due to transport costs, leaving large areas of the territory to be profoundly altered and abandoned. Disused production sites and quarries that are no longer in use are a threat to the landscape—they do not represent either the natural landscape or the cultural landscape but rather disturbing elements for both categories, which can, however, be transformed.

## 3. Method

### 3.1. General Premises

This work is part of a more general research path concerning the identification of ecologically oriented regeneration processes of the urban-building heritage of fragile territories [38–40]. It explores the spatial framework with reference to the four general categories of the natural, urban, social, and economic systems; each of them is observed in detail through different levels of depth [41,42].

This observation is structured according to a hierarchical scheme that organises the elementary spatial units (iUs) extracted from official sources. At the last level of the work breakdown structure (WBS) are the IUs that express more synthetic characteristics and capabilities that can be directly attributed to the four previous systems [43].

The result is the formation of a searchable information system for the calculation of complex indices to compare municipalities in the province by correlating different aspects—natural, urban, social, and economic—describing the territorial structure observed.

The aim of this knowledge framework is to support policies aimed at redressing the territorial handicaps resulting from the asymmetric development between coastal and inland areas, the latter often suffering from the conditions of inaccessibility of small centres located in inaccessible areas. In these areas, local communities still express a strong attachment to the territory and to the endogenous principles of sustainable development, places rich in natural, social and cultural resources, but at the same time facing significant risks of abandonment.

*3.2. Sources and Data*

The objective of 'Recovering Abandoned Landscapes' requires a reliable information base for comparing different information spatial units (iUs) to select similar or converging groups.

These relationships could be used to delineate spatial areas with similar problems and a similar potential for recovery.

Data sources are as follows:

1. With reference to the natural context, the Territorial Landscape Plan (PPTR) of the Region of Sardinia, the Geological and the Land Use Map from 2008, accessible on https://www.sardegnageoportale.it/ (accessed on 2 February 2023). The first shows the structure of the main systems with reference to the land aggregations that form homogeneous landscapes, while the two maps provide quantitative information on the abiotic and biotic components;

2. The database of the Italian National Statistics Institute (NSI) from 2011, with specific reference to the entire database on the details of the municipality and to the Census Section (CS) for the basic information on social, economic, and urban system aspects; the '8000 Censuses' sections, which offer an ordered selection of general databases for the purpose of presenting the phenomenon of social vulnerability; the 'Italy in detail' section, which includes the main municipal data of civil interest of the NSI (geographical, demographic, register, economic-productive, employment, patrimonial, etc.), important for the section dedicated to the urban system; the portals of the 'Italian Municipalities', which make it easier to consult data on demographic and income dynamics to describe the social system and aspects of income distribution; the Statistical Atlas of Municipalities (2011), with specific reference to ATECO (2007) data describing the production system of the territory;

3. The database of the Italian Real Estate Market Observatory (REMO)—by the Revenue Agency—provides general information on the structure of real estate; REMO data refer to the period from 2006 to 2022 and are aggregated in

   - the Microzones (MZ), consisting of five main classes aggregating 34 subsets distinguishing historical centres, semi-central and peripheral areas, industrial or craft areas, coastal and rural areas;
   - residential buildings, economic dwellings, boxes, industrial buildings, typical buildings, laboratories, warehouses, shops, offices, structured offices, villas, and cottages (by a topology);

4. The database of the Regional Plan of Mining Activities (PRAE), which lists the mining concessions and authorisations of the companies operating in Sardinia (updated to 2004): the PRAE also identifies active and inactive (i.e., upgraded or abandoned) quarries. This information tool has made it possible to measure both the degree of exploitation and anthropisation of the territory to produce traditional buildings and the possibilities for reactivating this production chain as an important part of local identities underlying a consolidated building culture.

The remaining data relating to climate, some production system specifications (i.e., for the agricultural system, PDO, and PGI products), and the wealth distribution index (Gini index) have been extracted from the platforms certifico.com and urbanindex.it (referring to 2018 and 2015, respectively).

### 3.3. The Model

The proposed model coordinates information units at different levels of description and evaluation. The information apparatus is structured in the form of a database in which the records (rows) are the territorial units (tU) that carry the information, and the fields (columns) are the attributes, i.e., the information units (iU). The attributes are divided into denoted and connotated; the former is of a mainly descriptive type, and the latter is of a mainly evaluative type. The latter are, in fact, representations of the value of everything from the point of view of the previously identified criteria that describe its capacity to be valuable.

#### 3.3.1. Territorial Units

The information base consists of three different databases (Figure 6):

1.  The first works at the municipal scale with the details of the NSI CS and consists of 5170 $tU_{NSI}CS$ and 140 $iU_{NSI}CS$ (the information units provided by the NSI database);

2.  The second consists of the areas identified by the MZs coming from the Real Estate Market Observatory (ReMO) of the National Revenue Agency (NRA), which extends to the 92 municipalities under study, forming a database of 3814 $tU_{ReMO}MZ$. Since the latter distinguish historical centres, zones of the first and second expansion, commercial and industrial zones, peripheral and coastal zones, and rural areas, through the functions of spatial association, the CS belonging to each MZ has been identified in order to be able to enrich the territorial real estate information (Real Estate Territorial Information) with the socio-economic and building information (Socio-economic Building Information) coming from the NSI at the retail of CS, thus being able to link heterogeneous levels of information related to different systems. This intermediate information apparatus consists of two sub-databases that coordinate a total of 5170 tUs and 1020 iUs (30 $iU_{NSI}CS$ for each of the 34 ReMO MZs);

3.  The third database consists of the Municipalities bounded by the NSI in 2023 and according to the 2011 survey. It uses the aggregations of the iUs attributed to each tUs of the two above-mentioned databases and elaborations made on the basis of the available thematic maps. The last database, which represents the entire set of observations and indices for all municipalities and is the main basis of the evaluation model, is composed of 92 $tU_{NSI}M$ (rows) and a total of 527 $iU_{NSI}M$ (columns).

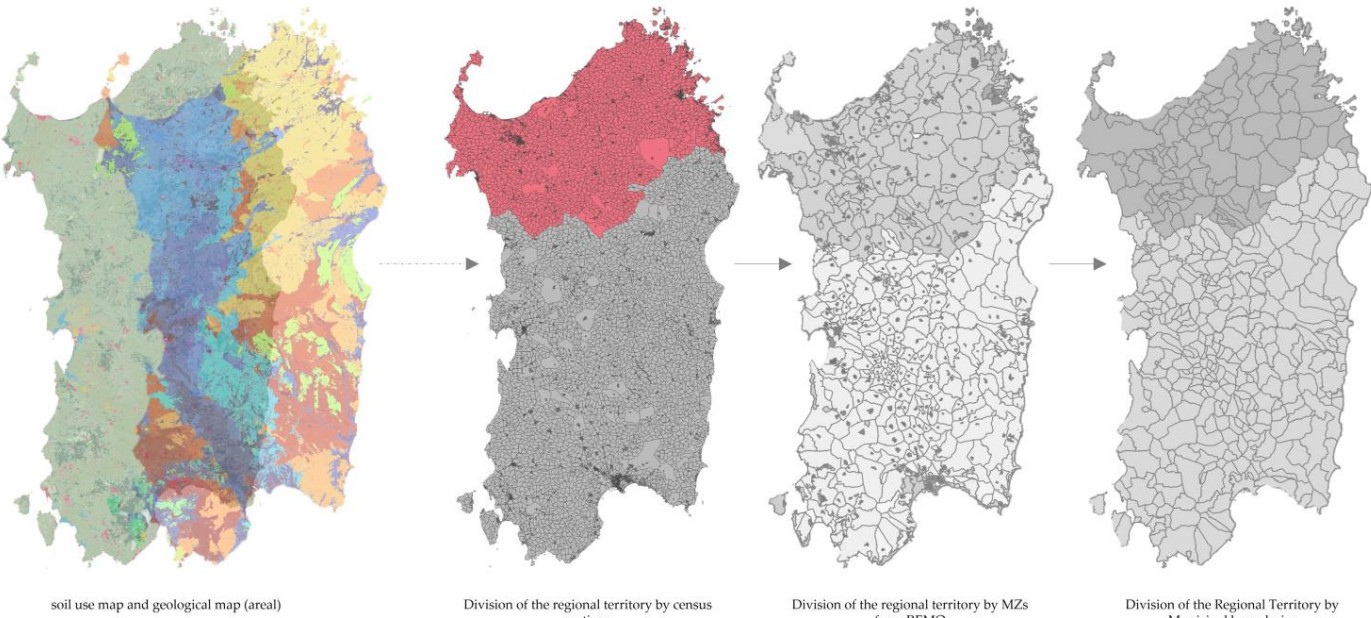

soil use map and geological map (areal)

Division of the regional territory by census sections

Division of the regional territory by MZs from REMO

Division of the Regional Territory by Municipal boundaries

**Figure 6.** Overlapping of available data in shapefile format and comparison between the different territorial units over the whole Sardinian regional territory. (Our processing).

### 3.3.2. The MAVT-Based Model

A model of multicriteria analysis based on the Multi-Attribute Value Theory (MAVT) [44–47] has been developed to characterise and compare the different main tUs on the basis of different value capacities, i.e., on the basis of different evaluation criteria. This capacity is expressed in the attitudes of the territories towards ecologically oriented redevelopment by each of the tUs that make it up.

The evaluation model used here is part of the operational research tools within the multidimensional analysis referred to by the tools that support shared decision-making processes, extending the knowledge of the utility sphere to the value sphere [48]. The first concerns mainly functional aspects, and the second also includes variables that refer more or less directly to ethical and aesthetic values [49].

The proposed model coordinates the cognitive process by means of a Work Breakdown Structure (WBS) [50,51], that is, a scheme of progressive unbundling of the main criteria in iUs that increasingly detailed information up to the information sources represented by different measures.

At all levels of aggregation, each iU is assigned a weighting factor that measures its importance in relation to those of the same group.

The WBS scheme is represented in the dendrogram of Figure 7. Each tU is characterised by a score that summarises its evaluation from the four points of view that outline its natural, urban, social, and economic profile.

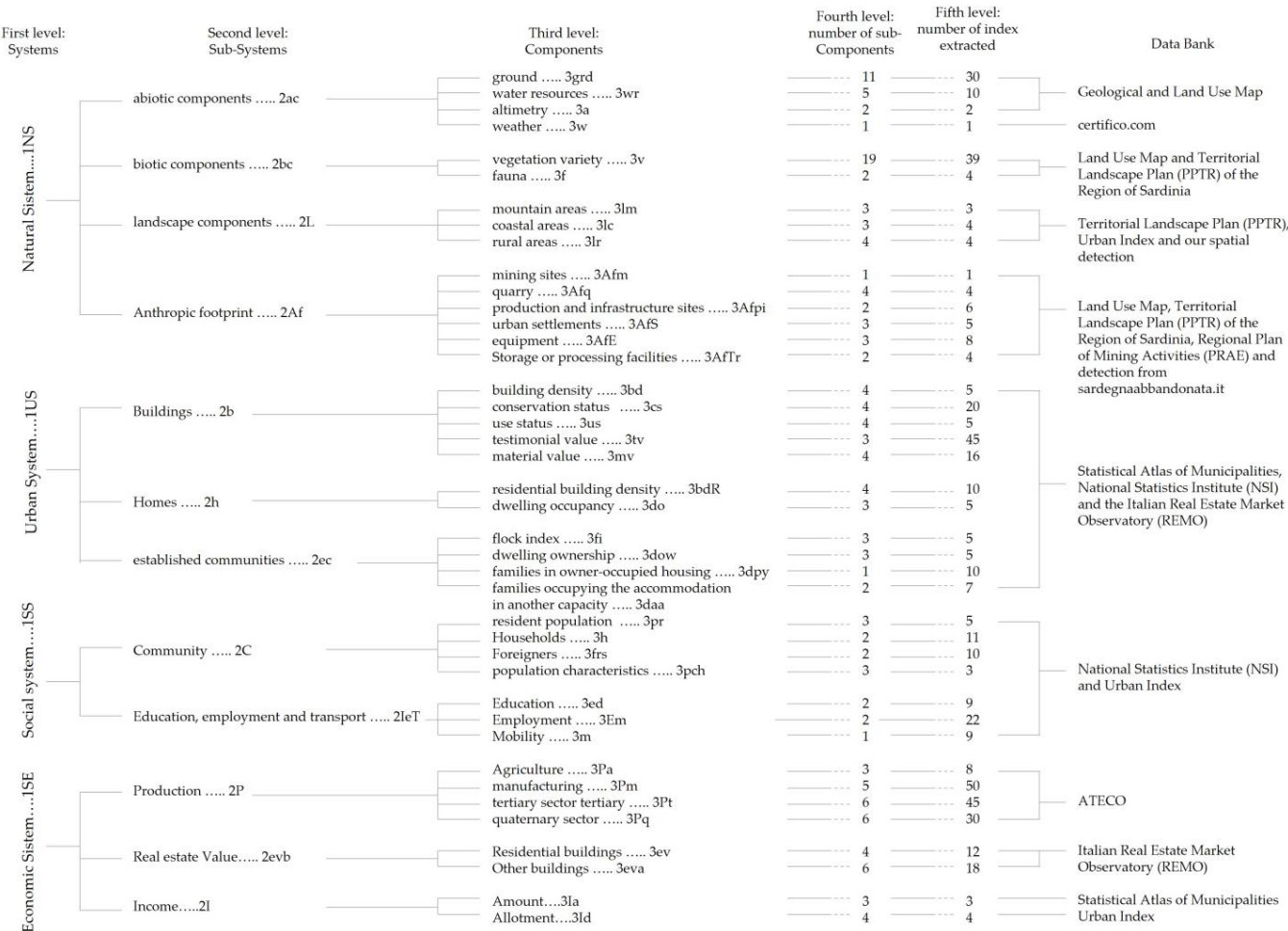

**Figure 7.** Dendrogram representing the distribution of the indices from the first to the third level of the WBS. (Our processing).

### 3.3.3. Attributes: Measurements and Valuations

In the proposed model, the attributes are the iUs that have a specific explanatory capacity at each of the WBS levels at which they are placed. The first level represents the value of the municipality from the point of view of each system. At the second level, the attribute expresses the aggregate value of the main components into which these systems can be disaggregated. Similarly, at the third level, the attribute expresses the sub-components and at the fourth level, the relative specifications for type and quality. Finally, the fifth level contains the measurements of the selected information units of the database from which the whole system is drawn.

The evaluation path starts from the level of maximum disaggregation, which contains the elementary ones. Each of them consists of the manifestation of a phenomenon expressed via a specific unit of measurement. At this level, the $k_{th}$ ($k = 1, 2, \ldots, m; m = 92$) tU is evaluated from the point of view of each $i_{th}$ ($i = 1, 2, \ldots, n; n = 527$) elementary attribute iU, by means of a value function $v_{ki}\left(\alpha_{iU_i}\right)$. This value is a score that varies for all evaluation functions in a range from 0 to 2, the minimum and maximum value, respectively; a score of 1 is a sufficient condition.

These functions increase or decrease depending on whether the observation concerns an event considered as a value or a disvalue.

The total value of each tU is then given by the sum of the products of all the attributes for the relative weights applied, starting from the elementary level up to that of maximum aggregation:

$$V_k = \sum_{i=1}^{n} w_i v_{iU_i} \tag{1}$$

where $w_i$ is the attributed weight of the $i_{th}$, and $v_{iU_i}$ is the score of the $k_{th}$ tU related to attribute $iU_i$.

For the application of Equation (1), it is necessary that

$$\sum_{i=1}^{n} w_i = 1 \tag{2}$$

The (1) refers to the aggregation of level 5 of the model where each elementary iU is transformed into a standard score and associated with it is an elementary weight. This elementary weight is calculated as the product of the weights at the higher levels, as follows:

$$w_i = w_h w_g w_f w_e \tag{3}$$

where $w_h$ is the weight of the corresponding ius group at level 4; $w_g$ is the weight of the corresponding iUs group at level 3; $w_f$ is the weight of the corresponding iUs group at level 2; $w_e$ is the weight of the corresponding iUs group at level 1.

This standard additive aggregation process is supported by the assumption of mutual independence of preferences between attributes.

### 3.4. Phases of Use of the Model

The model supports the critical analysis of the context studied in two phases: 1. identification of landscape-territorial vocations (landscapes); 2. definition of the main forms of landscape risk (abandonments); 3. identification of the main drivers of landscape risk and invariants of its resilience (drivers of abandonment).

1.  The first coordinates are the set of databases consulted in a hierarchical model in which each unit of information finds its place in each of the four systems mentioned above; on the basis of this information, detailed and synthetic index maps are produced in order to describe and consequently evaluate the municipal territories from the point of view of the four systems studied;

2. The second phase focuses on the aspects of abandonment by defining specific measurement indicators; the mapping of indices allows us to understand aspects of asymmetry between significantly different areas of the territory;

3. The third phase, based on the results of the previous two phases, describes the relationships between structural aspects (Natural System and Urban System) and anthropic aspects (Social System and Economic System) in an attempt to identify, measure, and represent landscape risk in terms of its main drivers.

## 4. Application and Results

### 4.1. Landscape(s)

4.1.1. Natural System

With reference to the above-mentioned definition of landscape risk, the natural system is examined here in terms of its physical components (abiotic and biotic), landscape structural values, and the extent and intensity of anthropogenic erosion (number of mining or production facilities or, more importantly, for environmental recharging, disposal, and processing plants, landfills, etc.).

The abiotic components are represented through the sub-components ground, elevation, water resources, and climate.

Considering the limitations of data availability and management, the aggregated indices of the soil sub-component summarise information on the presence of geotopes of greater or lesser historical and physical value, as well as the landscape and cultural uniqueness they represent [52].

Thus, the elementary iUs of the fifth level are aggregated to the fourth level by clusters of geotopes with homogeneous characteristics in terms of the amount of historical information they contain and/or the quality of their physico-chemical characteristics. The clusters are as follows:

1. Granitoid complex;
2. Rocks and haplitic groups;
3. Plateau basalts;
4. Sediments:

    4.1. Water bodies;
    4.2. Related to gravity;
    4.3. Historical succession.

These UIs, which are included in the fourth level of the dendrogram, are then aggregated to the *ground* component at the third level. In this case, weights are assigned based on the landscape and cultural potential identified in the geotope clusters. Thus, the value attributed to the diversity (and hence the richness) of the geotopes identified for each municipality at the fourth level is reversed at the third level, where specificity is more important than quantity: the fewer the different geotopes, the greater the value of their identity in terms of characterising both the landscape and the towns and cities that have been created and developed with the gradual consolidation of the technologies and techniques used for stone working.

The biotic components are divided into fauna and vegetation. The vegetation sub-component is described at the fourth level based on the grouping of the most common spontaneous or cultivated botanical species, considering their importance in terms of development patterns and lifestyles aimed at reconciling livelihood needs and biodiversity.

The landscape components refer to the sub-components of the coastal, mountain and rural areas, which are assessed according to their importance in terms of type and quality (i.e., in the case of coastal areas, the development of the coastline and the surface area of the municipality, as well as the distance from the urban centre to the coast).

The anthropic footprint is calculated based on the environmental footprint of anthropogenic—development patterns, such as production and urban settlements, infrastructure systems, large equipment, waste storage or transformation, and mining sites.

The resulting ranking places the municipality of Berchidda in the first place (Figure 8). Berchidda, similar to many other municipalities in Sardinia, is characterised by the relevance of the value of the natural system, as the score is strongly influenced by the reduced anthropisation. Only a few municipalities, including Esporlatu, Semestene, and Romana, have built-up areas that almost coincide with the boundaries of the whole agglomeration, and these are indeed at the bottom of the ranking.

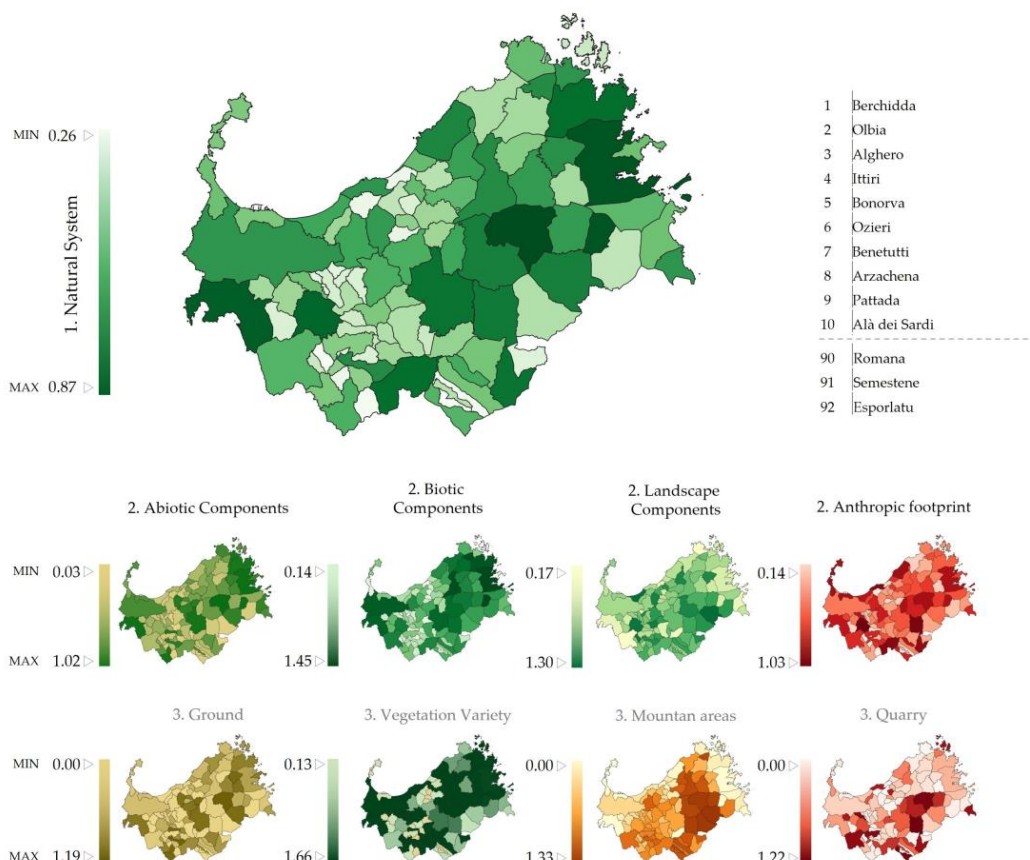

**Figure 8.** Map and ranking of the scores of the municipalities in the Natural System with mapping of the aggregate values at the second level and of certain sub-components at the third level considered most significant. (Our processing).

It is also noticeable that all coastal municipalities, apart from Olbia, Alghero, Arzachena, and Palau, generally score lower than inland ones. In this case, the landscape values are partly offset by the fact that large parts of the territory retain intact natural features of high value, which are sufficient to compensate for the anthropic imprint.

### 4.1.2. Urban System

The Urban System is interpreted as an open and organised system, defined in its forms by the flows of matter, people, and energy, from the interpolation of which its quality is derived [53].

To this end, the components of the urban system selected and identified here are the built heritage, the housing system, and the settled communities.

The built heritage (buildings) in its physical–quantitative (materials) and qualitative (culture) aspects is investigated through the sub-components that characterise it, such as building density, state of preservation, state of use, testimonial, and material value (referring respectively to the age of the buildings and the technological and material coherence of their formal apparatus).

Each of the sub-components aggregates its attributes with reference to the above-mentioned MZs del ReMO.

The set of attributes of the housing sub-component refers exclusively to the 'home' dimension (the space of the person); at the third level, this iU aggregates those present at the fourth, related to the above-mentioned MZs, as to housing density and occupancy rate.

The last component of the urban system is intended to describe the established communities (ideally the energy flows of the area) in quantitative (crowding index) and qualitative terms. From this point of view, the elements taken into consideration aim to assess the integrity of the identity of a community, representative of the territory to which it belongs. These elements are the family as a 'referent' and the house as a 'signifier', i.e., as an identity value that expresses precisely the degree of rootedness in the territory of origin.

The system's overall score (Figure 9) aggregates the three components, giving greater weight to the built heritage, valuing the physical and symbolic support with which communities identify themselves, and the relationship between owners/tenants for the significance highlighted above.

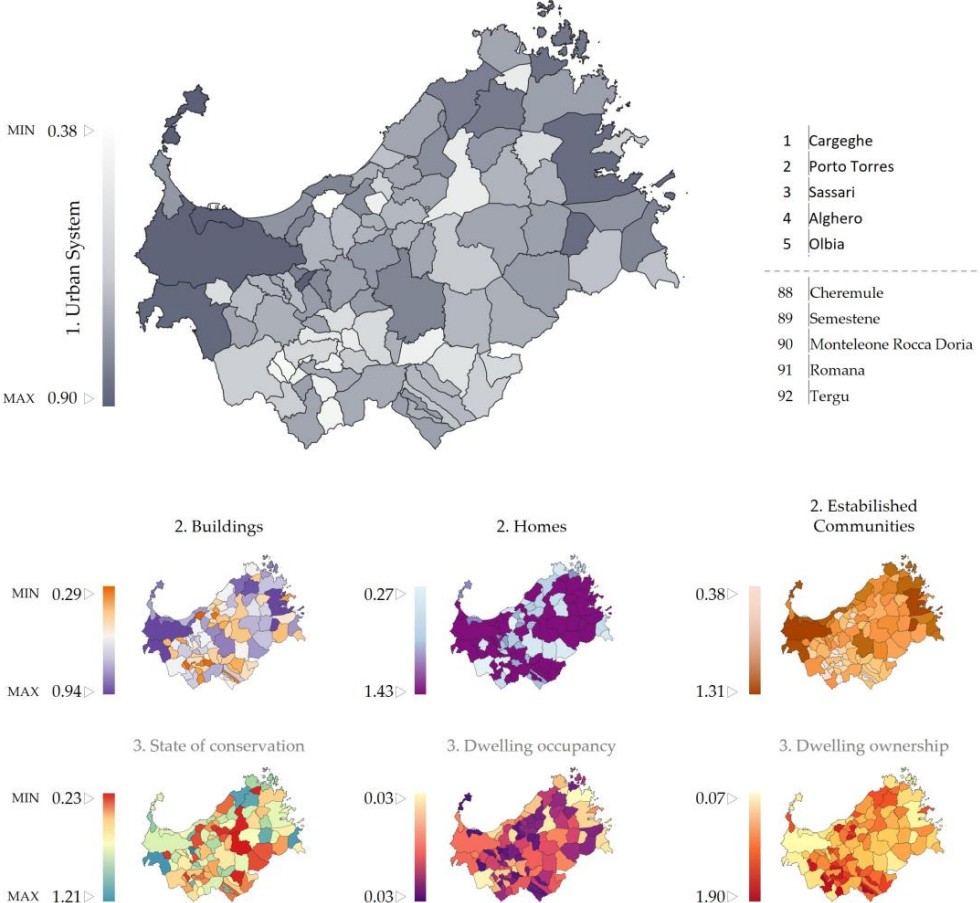

**Figure 9.** Map and ranking of the scores of the municipalities in the Urban System with mapping of the aggregate values at the second level and of certain sub-components at the third level considered most significant. (Our processing).

In a study focusing on the areas of northern Sardinia, the latter is of considerable importance. Except for Cargeghe, which is ranked first because of the historical and monumental value of the village and its good state of conservation, municipalities, such as Porto Torres, Stintino and La Maddalena, also rank high because they are among the few coastal centres not affected by significant processes of replacement of the original population.

The condition of inland municipalities is different, with an overall low level of building conservation and a high rate of building abandonment.

### 4.1.3. Social System

The components of the social system that are considered relevant here in order to represent the balance between the different territories relate, in general, to the way in which territorial wealth is more or less accessible to all.

Thus, the description of the communities settled in the municipalities under study aims at identifying the demographic dimension of the local social realities (communities) and the set of skills (employment and education) of the inhabitants.

In particular, in the communities component, the subcomponent of the resident population represents the demographic consistency of the settlement (population density) in relation to its greater or lesser attractiveness (intercensuary variation), while the household subcomponent represents a variety of aspects, among which those related to the risk of abandonment of areas were considered the most important; among these, the presence and consistency of family units (taken as a positive value), the presence of young single-parent families, elderly people living alone, etc. (taken as a negative value).

The heterogeneity of the sub-component (similar percentages of men and women, residents and non-residents, young and old) was considered important as a potential reserve of ability in view of the revitalisation processes of less vibrant urban centres.

The education and employment component, on the other hand, aggregates information on the level of education, employment and mobility services for the study and work of the resident population.

In terms of the overall score, Olbia, Ozieri, Pattada, Thiesi, and Sassari are in the top five (Figure 10).

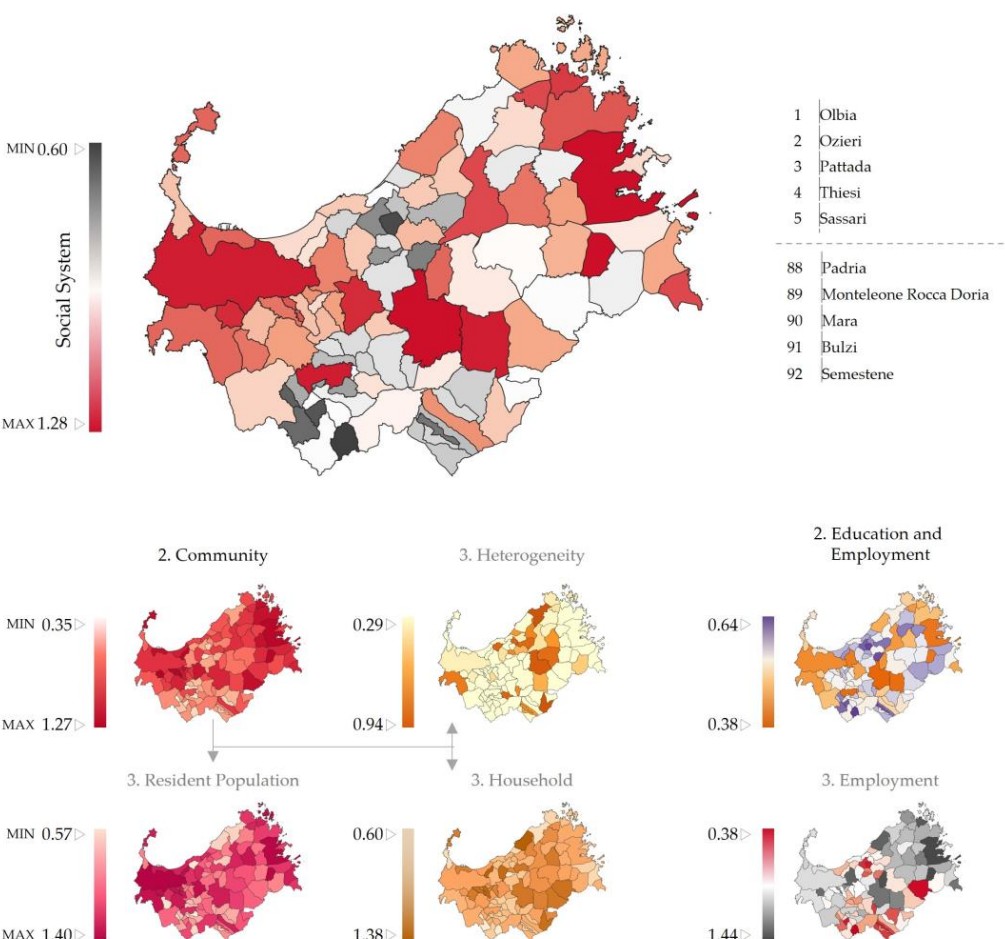

**Figure 10.** Map and ranking of the scores of the municipalities in the Social System with mapping of the aggregate values at the second level and of certain sub-components at the third level considered most significant. (Our processing).

Here, too, the score is conditioned by the size of the municipality and, thus, by the total number of inhabitants. The largest social realities are also those where economies of scale multiply, thus offering the greatest opportunities for development. As a result, they attract a heterogeneous population which, while enriching opportunities, reduces local identities and the sense of belonging.

4.1.4. Economic System

The economic system, in a general sense, represents the set of elements (actors, resources, technologies, knowledge) and their relations that give rise to the production of wealth, its exchange, use, and accumulation. The forms that this wealth takes in terms of satisfying individual and collective needs are described by the set of observations produced by the above-mentioned National Statistical Institutes, grouped into the three sub-components of production: the presence and size of entrepreneurial activity in the various sectors (primary, with specific reference to the extractive sector, secondary, and tertiary); the city (economic buildings value); the real estate sector and the people (income), in terms of personal income and its distribution.

The overall assessment places Olbia, Ozieri, and Sassari in the top two places, while Padria, Monteleone Rocca Doria, Mara, Bulzi, and Semestene are again in the lower part of the ranking (Figure 11).

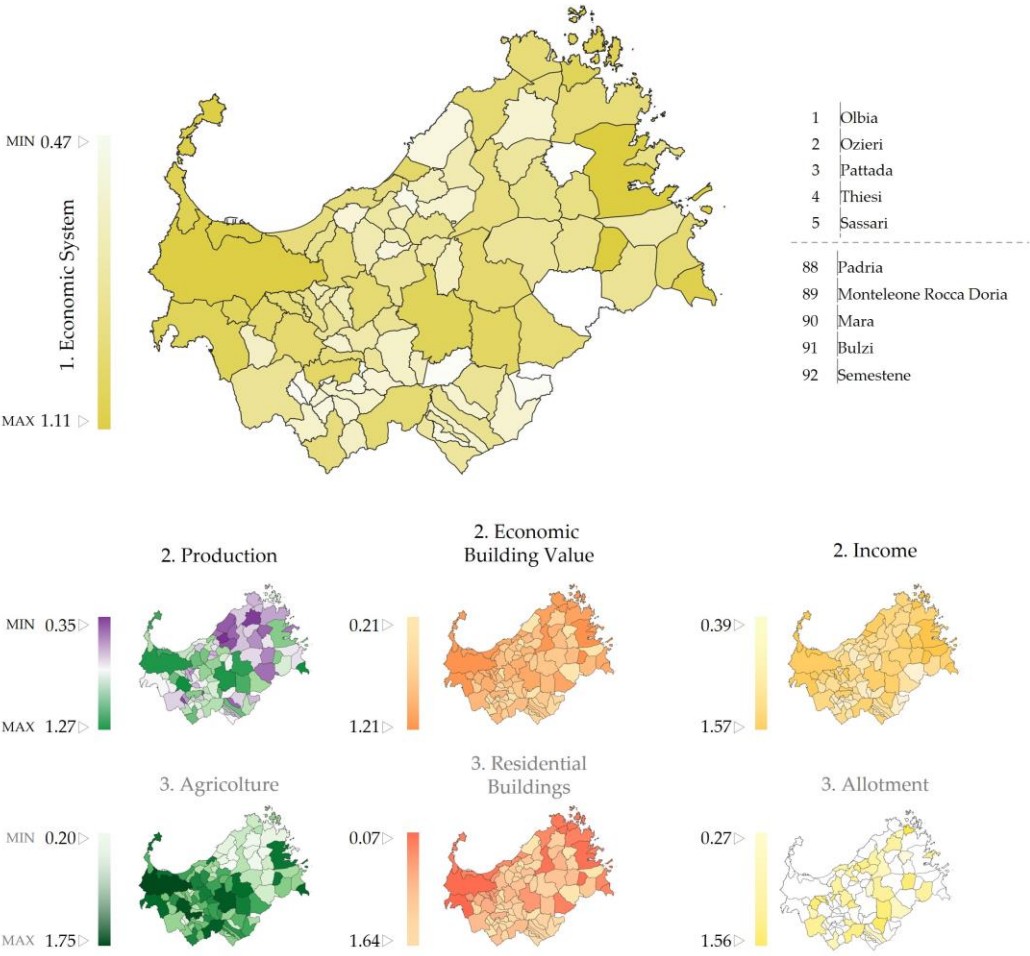

**Figure 11.** Map and ranking of the scores of the municipalities in the Economic System with mapping of the aggregate values at the second level and of certain sub-components at the third level considered most significant. (Our processing).

The production subsystem is more substantial on the western side, with the exception of Olbia and Budoni. The municipality of Olbia registers 8876 companies (5% of the regional total and 37% of the provincial total), with a higher growth rate than other areas of the region [54].

With regard to the value accumulated in the real estate sector, the survey highlights the disparity between the prices recorded by ReMo in the most prestigious coastal towns, particularly in the residential sector, and those in the inland municipalities. This disparity is due to the part of the property market that does not respond to internal or regional demand (or even to external demand in general) but is chosen by that part of the Italian and foreign population that is able to pay thirty times more for a property near the sea than in inland towns.

Confirming this, the survey shows that the amount of income, apart from the lower levels that characterise the flat inland areas, is basically the same between the coastal and mountain areas, suggesting that the value accumulated in luxury housing does not represent a share of the local income surplus, but rather that this asset is a container for the excess liquidity generated elsewhere.

### 4.2. Abandonment(s)

Due to its heterogeneity, landscape risk is presented according to the different forms of land abandonment. The abandonment is considered according to the two main dimensions of local identity, the collective and the individual. The former presents the phenomenon in its entirety with reference to economic and urban systems, and the latter with reference to the residential and family dimension.

### 4.2.1. The Collective Dimension: Land Abandonment

As explained in the Materials chapter, the causes of demographic decline are mainly to be found in the economic substitution of the original productive dimensions by the markets of the new modernity. The forms of land abandonment identified, therefore, concern the following:

1. Stone economies, given by the ratio between the total area of inactive quarries and the total area of active quarries within the boundaries of each municipality;
2. Agricultural economies, given by the ratio of the inverse of the agricultural area in use to the total agricultural area;
3. Industrial settlements, given by the number of disused industrial buildings (taken from the website sardegnaabbandonata.it [55]) out of the total municipal territory;
4. Built heritage, given by the total number of unused buildings out of the total number of buildings.

From the analysis carried out, it can be seen that the condition of abandonment of the territory mainly affects the inland areas, with the exception of the municipalities of Golfo Aranci and Palau, which, although they are poles of tourist attraction, do not have significant manufacturing structures and activities; similarly, the two municipalities of Trinità D'Agultu e Vignola and Badesi, precisely because they belong to the Gallura district, fully express the typical features common to the inland areas despite the significant landscape value of their waterfronts (Figure 12).

The thematic maps describing the dimension of abandonment in relation to the four economic systems mentioned above show the profound transformation of the economy of many centres in northern Sardinia, which is becoming increasingly tertiarised, confirmed by generalised population growth.

The demographic dynamic that has affected the whole of Sardinia, with a marked and constant increase from the fifties to the eighties and a subsequent slowdown until the first decade of this century, is reflected, to a greater or lesser extent, with due differences, in all the municipalities mentioned above, which moreover represent the dynamics of recent decades.

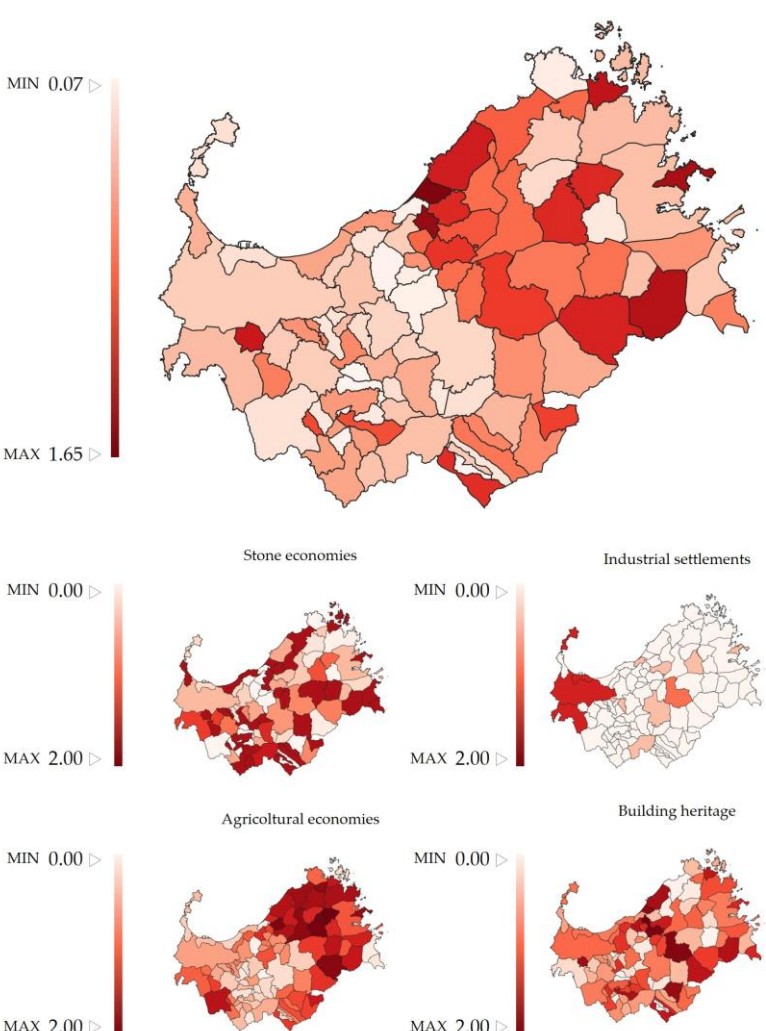

**Figure 12.** Mapping of the forms of abandonment concerning the collective dimension with a histogram by classes of abandonment of the built heritage in coastal and inland areas. (Our processing).

4.2.2. The Individual Dimension: Home Abandonment

The dimension of home abandonment was investigated with reference to:

1. Dwellings—disused ones compared to total dwellings;
2. Historic centres—residential buildings not in use in historic centres in relation to the total;
3. Dwelling ownership—rented or otherwise occupied dwellings compared with owner-occupied ones;
4. Thinning—the number of residents per dwelling.

The resulting map is almost complementary to the previous one and shows that the lowest rates of utilisation of the housing stock are in the above-mentioned Aglientu, Trinità d'Agultu e Vignola, Badesi, Golfo Aranci, Palau, Stintino, and in the other municipalities on the east coast (Arzachena, Loiri Porto San Paolo, San Teodoro, and Budoni), except for Monteleone Rocca Doria (Figure 13). In these municipalities, with the exception of Monteleone Rocca Doria, the population is growing, with a change in lifestyle, a better redistribution of the urban population between historical centres and expansion areas, and a more organic relationship between owned and rented housing.

**Figure 13.** Mapping of the forms of abandonment in relation to each dimension, with histograms by classes of the degree of thinning and occupancy of dwellings in coastal and inland areas. (Our processing).

In particular, the ratio between the number of inhabitants and the number of dwellings does not deviate from the Italian average (1.89) of the recording values of 1.34, 1.67, and 1.93 for the three quartiles calculated.

In order to better understand the above, the demographic dynamics and property prices of the above municipalities have also been examined in comparison with other municipalities that are considered to be useful points of comparison.

Figure 14a shows the population dynamics that highlight the apparent contradiction between population growth and the state of low housing use. An overview of the entire province of Sassari from the point of view of real estate quotations (Figure 14b) highlights the considerable gap between the high number of municipalities (67), in which average real estate quotations fluctuate between 525 and 920 €/sq.m., and the extreme variability of quotations among the remaining 25 where quotations vary from 920 to 16,000 €/sq.m. This disproportion indicates the impact on the housing economies of the tourist accommodation sector due to the exceptional characteristics of the Sardinian coastal landscape. This asymmetry has, therefore, led to a complete reinterpretation of the living experience and the relationship between home ownership and use.

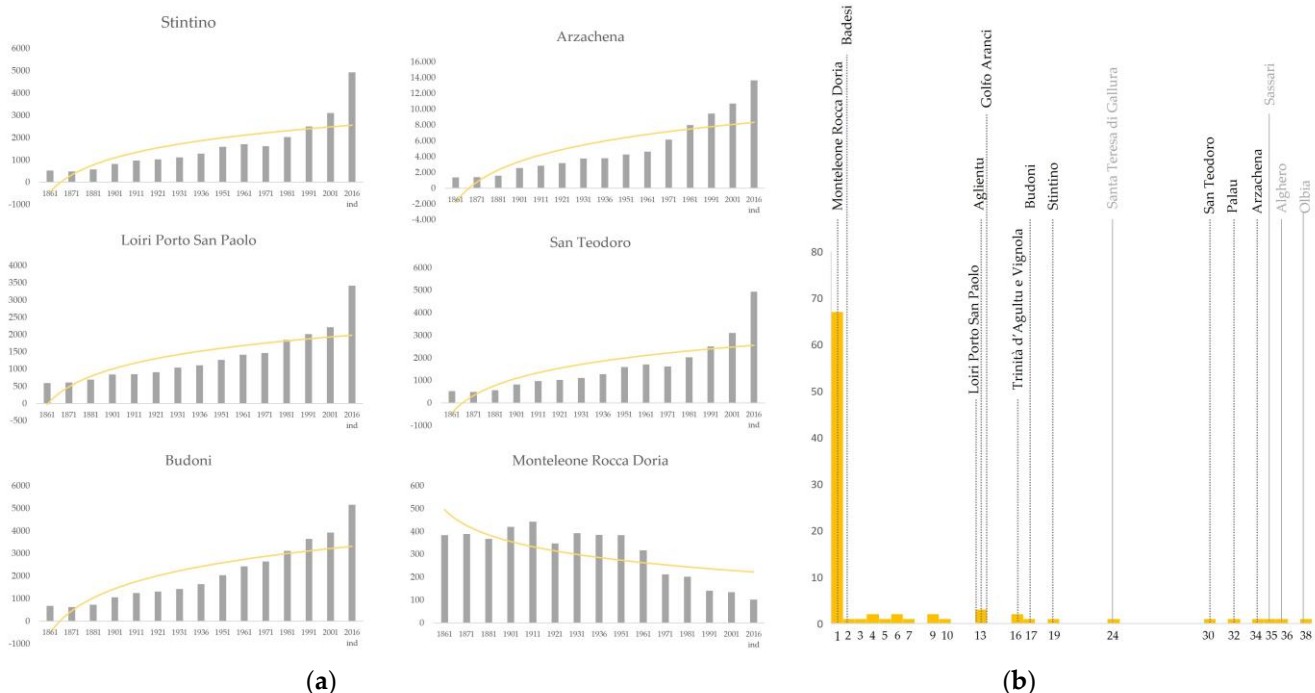

(**a**)                                                   (**b**)

**Figure 14.** (**a**) Demographic evolution in the towns mostly representing the home abandonment; *x*-axis, years; *y*-axis, population; (**b**) histogram with classification of economic values of dwellings; *x*-axis, real estate prices classes, from the lowest class (525–919 Euro/sq.m) to the highest (15,902–16,295 Euro/sq.m); *y*-axis, number of municipalities per class. (Our processing).

### 4.3. Interpretation: Drivers of Abandonment

This final part uses the extensive analysis carried out in the previous phases to support considerations aimed at identifying the drivers of abandonment between the iUs, built through the proposed model and attributable to the four systems.

To identify the different declinations of the degree of abandonment in a heterogeneous territorial context, the findings of the two previous analyses were compared with the main drivers of abandonment taken here as the main dimension of landscape risk.

These analyses and the resulting interpretations have revealed a dual register of the phenomenon of abandonment, which affects inland and coastal areas differently, showing, in some cases, different and sometimes divergent trends.

The drivers were selected from the iUs of the different levels of the analysis on the Urban Social and Economic system and with reference to the two different dimensions, collective and individual, described above.

With regard to the collective dimension of abandonment, i.e., the phenomenon caused by general structural changes, this study looked at the following:

- The productive system, in terms of the relationship between the causes and effects of land abandonment (quarries, agricultural land). The greater efficiency in the use of the productive potential of local assets indicates a certain resistance to abandonment in both clusters, but more so in the coastal areas, both because of their greater dynamism and because of the location of some important productive settlements (Figure 15a);
- The landscape components: although with a very low index of determination, both coastal and mountain communities show an increasing tendency to land abandonment. Because of the value of the landscape, communities seem to opt for the possibility of exploiting local resources (Figure 15b);
- The characteristics of settled communities in terms of density, heterogeneity, and occupation of dwellings. In this case, the two clusters interpret the relationship between the abandonment index and social integration differently. Coastal areas

are, on average, more socially integrated and seem to resist abandonment, unlike mountain communities (Figure 15c).

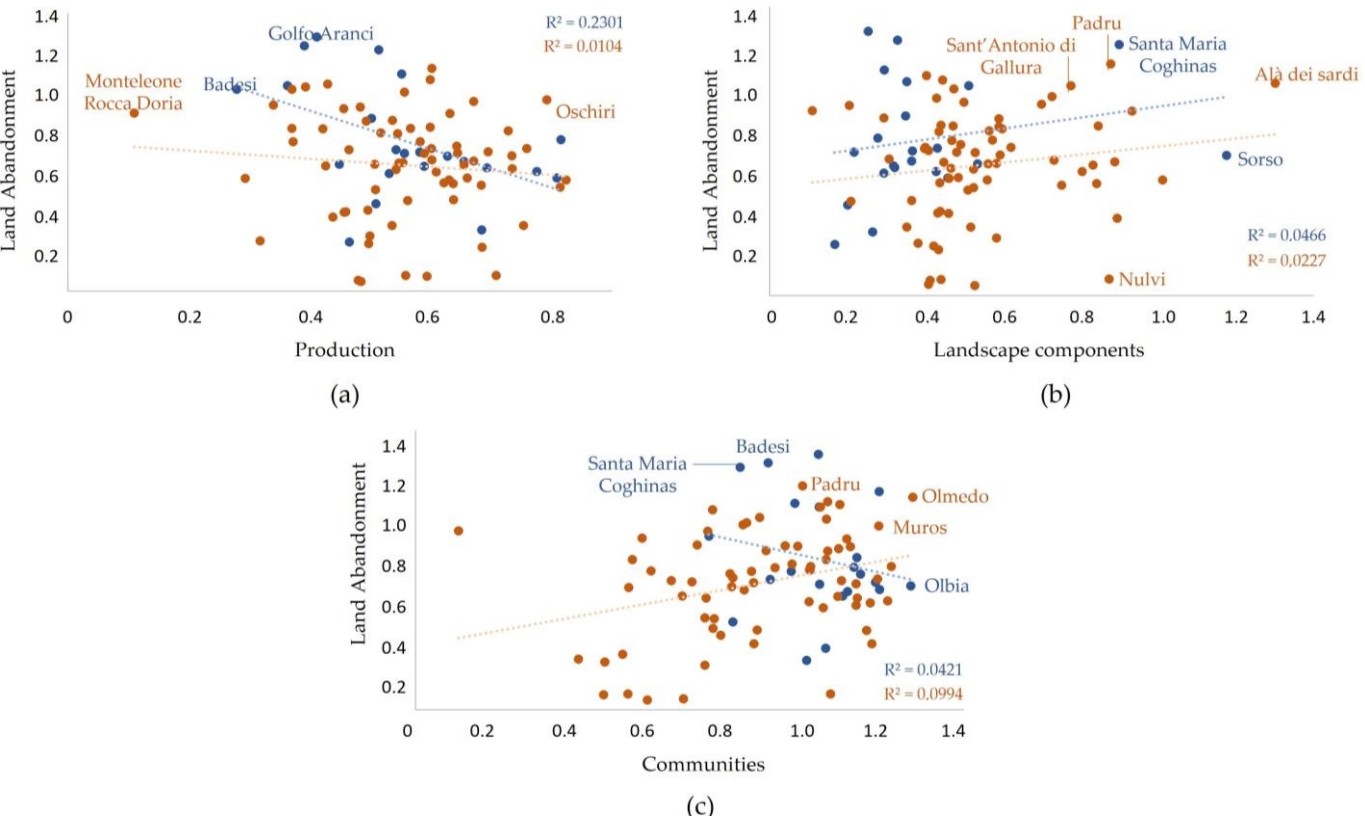

**Figure 15.** Correlations between the index of land abandonment and the drivers of the structure of the production apparatus (**a**), the drivers of landscape components (**b**) and the value of the characteristics of settled communities (**c**) in coastal areas (blue) and inland areas (orange). (Our processing).

Regarding the individual dimension of abandonment, the identified drivers are as follows:

- The characteristics of the urban housing system in terms of the state of conservation and the heritage value of the buildings. In mountain areas, the degree of abandonment is higher for buildings in a poor state of preservation, while in coastal areas, on the contrary, the tendency to abandon houses increases for those in a very good state of preservation due to the seasonal use of the buildings. The heritage value constitutes a driver of resilience in both clusters, showing how urban identity tends to create a stable link between the population and the house (Figure 16a);
- The economy of the settled communities in terms of income and employment levels. The two clusters are very different in these two respects; the richer coastal areas confirm one of the main risks of abandonment due to the friction between the traditional economy of local production and the newer economy of services linked to seasonal tourism; in fact, while the richer inland communities maintain the link with housing, the opposite is the case in the coastal areas, precisely because of the preponderance of housing used by non-residents (Figure 16b);
- The characteristics of households in terms of the number of members and the relationship between households and housing are measured by the percentage of owner-occupied dwellings. For both clusters, the size of the household is an element of resilience that is more evident in the interior; on the contrary, the relationship between the abandonment rate and the number of households living in rented accommodation outlines divergent trends in the two clusters; in coastal areas, the tendency towards

abandonment denotes those cities where the high number of rented households in relation to the total is justified by the low number of inhabitants and high market prices of dwellings, as well as the presence of temporary work and study opportunities; in inland areas, the resilience to abandonment denotes a more complex urban reality, as in Cargeghe and Luogosanto, where a wider range of economic activities and more job opportunities seem to incentivise renting (Figure 16c).

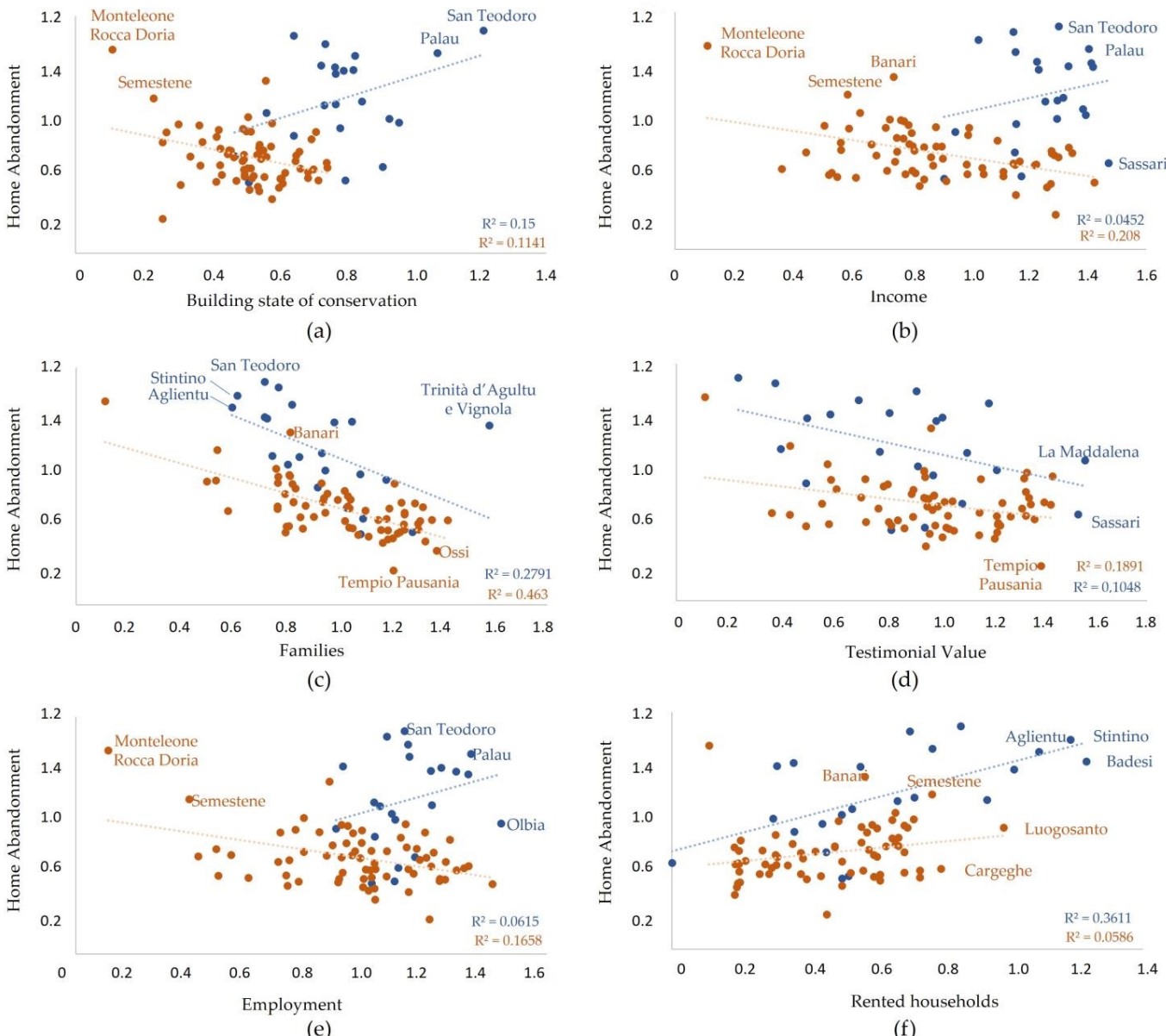

**Figure 16.** Correlations between the index of home abandonment and its drivers with specific reference to the state of preservation of buildings (**a**) and their testimonial value (**d**), the income of settled communities (**b**), their level of occupation (**e**), the composition of households (**c**), and the number of households in rented flats (**f**) in coastal areas (blue) and inland areas (orange). (Our processing).

## 5. Discussion

The research is part of the studies and policies for the revitalisation of disadvantaged territories that characterise both national and European territories [56–60].

In Sardinia, scientific interest seems to focus on the demographic malaise (SMD) and the difficulty in accessing basic services (DAS) [61,62], similar to Italy, where studies focus sectorially on depopulation—interpreted only in quantitative terms—[63], on the poor management of the economic resources of the 'public machine' to effectively support development projects [64] or even on the insufficient access to services that characterise inland areas [65].

In Europe, many countries have promoted policies aimed at territorial rebalancing and the identification of specific funding to support the improvement of the quality of life in small towns, such as in France with the 'Revitalisation des centres-bourges' and the 'Petites villes de demain' promoted in 2014 and 2020, respectively [66,67], or geographical areas (mountain, rural and border) with specific development needs [68].

The proposed study prefigures a broadening of the scientific and operational interest in territorial disadvantage, according to a further meaning of abandonment that in a holistic vision of the components that build the landscape can lead to more articulated solutions and directions, with respect to a very complex reality [69].

The first phase provided integrated knowledge aimed at identifying potentialities and criticalities of both the original physical conformation (geological, water etc.) and cultural transformations (productive, urban, etc. settlements) of the analysed context. The articulation of the abandonment concept, addressed in the second phase, constituted one of the main focuses of this research, declined with reference to the two dimensions of associated life, the home and the territory.

These are represented by specific indices measuring eight different aspects of abandonment identified as a result of the observations in the previous phase. The third phase defined the incidence of the drivers of abandonment by highlighting the different declinations of landscape risk in a heterogeneous territorial context; one of the two interpretations refers to the manifestations of impoverishment and territorial disadvantage characterising the hinterland; the other refers to the overlapping of the real estate and service sectors on a barely cohesive productive economy.

The results obtained are in keeping with the representations produced by the National Atlas of Rural Territory [70], which, with specific reference to the areas of northern Sardinia, identifies marginal areas as affected by differentiated criticalities that induce the abandonment of homes and towns, tourist areas, and areas in agricultural decline that induce land abandonment. Several studies on the national territory have highlighted this dichotomy between marginalisation and development [71].

The sharability of this conoscitive and evaluative path consists in the application of a model based on a WBS, easily traceable top–down and bottom–up, whose criteria are ordered in a structure articulated in four systems (natural, urban, economic, and social). The methodology and tools used make the model flexible and extendable to other contexts, as well as integrate additional data to complete the knowledge dashboard [72].

The limitations of the proposed model include, for example, the lack of useful data to understand the physical–geographical constraints of the territories (e.g., orographic analysis), updated dataset coming also from other institutions [73], and network analysis with reference to the measurement of accessibility [74,75].

Since these data need to be updated and contextualised with respect to the proposed articulation of the criteria, we will postpone a more detailed analysis to a monothematic study on accessibility.

Another aspect that makes it difficult to immediately understand the results of the proposed mappings consists in the heterogeneity of the main tUs in terms of size and density (surface area, population, the extent of the built heritage, etc.), which is combined with the many salient features of these territorial units, in particular, the prevailing natural, urban, economic, and social characterisation.

The general perspective of the presented study is to extend the methodology applied to the entire regional territory in order to

- support wide area planning in identifying development strategies that originate predominantly from their own territorial capacities in order to foster forms of autonomy consistent with the profile outlined by the measures implemented against the COVID-19 pandemic [76];
- address territorial rebalancing strategies that can mend the dichotomy between inland and coastal areas [77–80];
- reinterpret abandoned industrial, productive, and urban sites by recognising them as resources with a high capacity to generate new development paths for local communities, according to a holistic vision that integrates environment, society, and economy, as proposed by the new Regional Development Plan (RDP) 2020–2024 [81–83] and also in line with the objectives of the SDGs [84].

## 6. Conclusions

The aim of this study was to provide a framework for observing and interpreting landscape risk at the provincial level, with reference to the phenomenon of abandonment of the most important territorial components in northern Sardinia. The complexity of the causes of abandonment and the heterogeneity that characterises the various territorial units that make up the study area made it necessary to construct a structured information model with reference to the natural, urban, economic, and social systems. The model supports an initial overall assessment of the studied context, as well as a subsequent differentiated evaluation by clusters of municipalities (inland and coastal areas) in which the two dimensions of abandonment (land and home) were represented.

The model is a cross-sectional query tool of a database, the contents of which are also evaluation indices built for the specific purpose of representing abandonment. The context studied, consisting of 92 tUs (municipalities), is characterised by 527 iUs between observations and indices, grouped with reference to the main territorial systems.

Through a differentiated analysis of the two clusters of municipalities, this study has highlighted convergent and divergent trends that reveal, on the one hand, the profound transformation of coastal areas and, on the other hand, elements of the resilience of inland areas, which could be useful to drive future large area planning. Coastal areas show elements of risk of landscape abandonment due to trends typical of areas where the link between the agricultural economy and local production and the housing system is interrupted by the emergence of transitory service economies. The former is characterised by long-term investments and slow processes of mutual adaptation, the latter by sudden and irreversible transformations due to tourism development. On the contrary, inland areas manifest the typical features of local communities characterised by a strong connection to places where economic and family structures tend to retain the population and keep alive the housing and economic heritage consolidated by the combination of the characteristics of the land and the organisation of work.

**Author Contributions:** Conceptualization, A.M.S. and C.C.; methodology, C.C.; software, C.C.; validation, C.C.; formal analysis, C.C.; investigation, C.C.; resources, A.M.S. and C.C.; data curation, C.C.; writing—original draft preparation, C.C. and A.M.S.; writing—review and editing, A.M.S. and C.C., visualisation, C.C.; supervision, A.M.S.; project administration, C.C. All authors have read and agreed to the published version of the manuscript.

**Funding:** This research received no external funding.

**Data Availability Statement:** Not applicable.

**Conflicts of Interest:** The authors declare no conflict of interest.

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
