# Peer review of "The Realms of Abandonment: Measures and Interpretations of Landscape Value/Risk in Northern Sardinia (Italy)"

_land, doi:10.3390/land12071274_

Round 1
Reviewer 1 Report
1 1. “Over the last century, human activities have degraded 75 per cent of 84 the planet's habitable land, and at current rates this will rise to 90 per cent by 2050.” What does it mean? In what terms? Does the reference 18 apply to the whole section?
2 2. Figure 5. There should be agriculture instead of agricolture.
3 3. Figure 6 is hardly understandable. The letters are too small (applies to all figures).
4 4. Lines 245-246: “…leaving large areas of the territory to be profoundly altered and abandoned, conditions that today threaten landscapes that are no longer natural.” Not understandable. Seems to be the opposite. If the landscapes get abandoned, they start to naturalize.
5 5. Line 283: One of the two 2s is unnecessary.
6 6. Line 341: “the most significant iUs of the first two 341 databases”. What criteria have been applied to identify them?
7 7. Figure 8. The data structuring is questionable. 3AfS urban settlements are not an element of natural system 1NS but urban system 1US. Established communities 2c relate to community 2c too. Economic value 2evb consists not only of the value of buildings but other assets too, e.g. agricultural or mining ones. Please try to reflect these relationships in the diagram that may be more sophisticated.
8 8. Line 466. I would exchange the word “unsustainable” to “anthropogenic”.
9 9. Figure 8 - Established communities 2ec criteria consist of some qualitative only data that seems not to suite to the assumed further (line 499 on) methodological idea: “The last component of the urban system is intended to describe the estabilished communities (ideally the energy flows of the area) in quantitative (crowding index) and qualitative terms.”
1 10. In lines 401-409 you talk about weights. I can’t find which weights have been assigned to distinct qualities. E.g. how do you compare the significance of education and material value of buildings or dwelling ownership?
1 11. Lines 605-606. You write that “From the analysis carried out, it can be seen that abandonment is more prevalent in inland areas (Figure 13).” However it is not evident, judging on that figure. To draw such conclusions you should apply some spatial distribution measurement methods.
1 12. The Figure 14 is not clear. It should be described in a better way and translated to English. The same Fig. 18.
1 13. Lines 641-644 – the sentence is difficult to understand.
1 14. Figure 16: labels denoting variables in the charts are missing that makes them hardly understandable. Please consider it elsewhere too.
15. “Figura 18.” The number of households in rented flats is f not d.
Author Response
Dear reviewer,
cover letter on changes to the manuscript is attached.
Thank you very much for your valuable comments and suggestion.
Best regards.

Reviewer 2 Report
A manuscript entitled 'The realms of abandonment submitted for review. Measures and interpretations of landscape Value/Risk in Northern Sardinia (Italy)' concerns the possibility of building a database containing a number of parameters regarding the condition of the main elements of the territory of northern Sardinia. The Authors have chosen an interesting area of research from the point of view of geography and significant from the point of view of cultural heritage and socio-economic changes, which positively affects the significance of the manuscript. The synergy database proposed by the Authors may describe to some extent the state of vitality/abandonment of urban units in relation to the natural, urban and socio-cultural components. The manuscript focuses on applicability in the area of northern Sardinia, however, the database constructed in this way also shows applicability for the entire area of Sardinia.
In the research, the Authors use the multicriteria analysis model based on the Multi Attribute Value Theory and selected parameters for identification of landscape-territorial vocations (Landscapes) and definition of the main forms of landscape risk.
Dear Authors,
I hope that my comments and suggestions below will help you to improve the manuscript.
Major comments:
I suggest the Authors consider changing the topic of the manuscript. In my opinion it does not fully correspond to the content of the manuscript. If the title of the manuscript is to include ‘Measures and interpretations of landscape Value/Risk’ in the manuscript, authors should pay more attention to other measures of landscape value and their description in the introduction. It is also not uncommon to see a topic composed of two sentences separated by a period. The main emphasis of the discussion was devoted by the Authors to the drivers abandonment, which also leads to suggesting a change of topic.
On the surface, the manuscript has the correct layout, but after careful analysis, I found it to be too elaborate. The manuscript in its current form may not enjoy much interest due to the degree of development and graphic presentation. I think the Authors should focus on shortening the manuscript. At the same time, they should avoid commonly known facts and emphasize the conclusions drawn from their results. The Authors conducted research that allowed them to diagnose the phenomenon of land and home abandonment, however, in my opinion, they develop side plots in too much detail in the manuscript. For example, an extensive description of the geological issues of the entire region.
The Authors used 63 sources (including 2 self-citations). This is a large number for a research paper. I propose to resign from some of the items, especially items in national languages and references that mutually duplicate the quoted facts, or some of the older references. At the same time, in my opinion, the article lacks a reference to sources regarding changes in land cover and land use as a resultant measure of socio-economic changes in metropolitan areas or functional urban areas (urban core and its surroundings) in Europe.
In the case of using such a common method as the multicriteria analysis model based on the Multi Attribute Value Theory, I see the Authors' contribution only in the proposed set of parameters and the results achieved for the region of northern Sardinia. Authors must highlight the strengths of their results. At the same time, the article does not discuss the weaknesses of the authors' research (the database used and the model built), especially the emphasis on the factors that may affect the conclusions drawn, or the shortcomings of the proposed database. For example, from my point of view, there is no analysis of the density of the communication network in the analyzed areas, as well as the degree of diversification of their area, or finally a critical approach to the reference units for which the data and results used have been compiled.
Authors must necessarily in the case of section 3.2. 'Sources and data' specify the time frame of the data used. This is an important but omitted aspect of the manuscript. Data and their changes are always related to a moment or time period.
The Work Break-down Structure (WBS) scheme is represented in the dendogram of Figure 8 (lvl 1 to 3), but in lines 378-380 is mentionet about fourth and fifth level. The existence of levels four and five should be indicated in Figure 8.
In my opinion, the presented figures also need improvement. General note for Figures, legends need to be corrected. Description of max and min loses information. I advise against using a continuous color ramp for choropleth maps, it is more advantageous to use a discrete color ramp and divide objects into groups with an assigned range of values.
In the current form, the analysis of the presented figures is difficult because they are illegible (for example Fig. 2, 3, 4, 6). I recommend assigning the analyzed administrative units identifiers that will consistently appear on choropleth maps, and in the text of the table with the names of administrative units and identifiers assigned to them. I suggest to present only the research area in the figures, showing the whole of Sardinia reduces the readability of the figures (e.g. Fig. 7). In addition, I recommend considering whether all the pieces are needed. For example, the map of the provincial municipalities to be analyzed and ranking of the population per municipality in Sassari can be presented together on one choropleth map with the mentioned identifiers and not names (while part b Fig. 6 does not provide significant information, it can be replaced with a short conclusion in the text of the manuscript).
In addition, the Authors should consider whether presenting the first 30 positions in each ranking makes sense (Fig. 9-12), if their detailed discussion is not included in the text. It is better to focus on the identification and detailed presentation/discussion of the obtained extreme values, which will allow for better identification of the issues raised in the manuscript and discussion of the driving forces discussed. In my opinion, demographic evolution in the towns (Fig. 14) in its current form adds nothing to the manuscript, the graphs reproduce practically the same scenario. I suggest dividing cities into groups based on the value of the land abandonment and present demographic evolution together for cities from the same group of the land abandonment values.
The Authors conducted a specific discussion of their research results because they do not discuss them with the achievements of other authors devoted to other regions or countries. In 'Discussion: drivers of abandonment' section, there is not a single reference to research from other regions or countries. This must be corrected. This must be corrected or this section should be renamed. Discussion elements, on the other hand, appear in the 'Conclusions' section. I request the Authors to systematize the content of the manuscript and separate a section with a Discussion of the results and a short Conclusion containing the most important facts resulting from the research and generalizations as well as plans for the future.
Minor comments:
Please carefully follow the use of abbreviations. I have observed that some abbreviations are entered several times in the text or are entered in the wrong place (for e.g. line 56 'United Nations' next line 100 'United Nations (UN)' etc.)
Line 84-85 requires a data source.
Line 285 'National Statistics Institute (NSI)' such a statement is confusing to the reader from another country, it is better to specify which country or region this statistical office is.
Line 181 ‘at both national and European level, this phenomenon is not so 181 much a demographic decline as a movement of the population from inland areas to large 182 metropolitan centers [33]’. Item [33] is from 2015. Please additionally refer to, for example, data from EUROSTAT for the current state.
Author Response

(The authors gave the same response as above.)

Reviewer 3 Report
The article meets all the criteria of scientific papers. The only suggestion is to indicate whether the research results are consistent with other analyzes in this area.
The article shows the relationship between the abandonment of settlement units in rural areas and landscape value. First of all, it illustrates the methodology of studying this relationship, taking into account the data set and GIS tools.The topic is extremely important because the phenomenon under study is becoming common in Europe. It is necessary to be able to identify this process in order to remedy it and not allow it to develop.First of all, it defines the research methodology using GIS - a modern tool for spatial analysis.There is no need to improve the methodology in the article. You may consider testing it for other areas.The conclusions consistent with the evidence and arguments presented and they address the main question posed.The references are appropriate, although works on countries other than Italy may be included.Tables and figures are prepared correctly.
Author Response

(The authors gave the same response as above.)
